# Intranasal Resveratrol Nanoparticles Enhance Neuroprotection in a Model of Multiple Sclerosis

**DOI:** 10.3390/ijms25074047

**Published:** 2024-04-05

**Authors:** Ehtesham Shamsher, Reas S. Khan, Benjamin M. Davis, Kimberly Dine, Vy Luong, M. Francesca Cordeiro, Kenneth S. Shindler

**Affiliations:** 1Institute of Ophthalmology, University College London, London EC1V 9EL, UK; ehtesham.shamsher.16@alumni.ucl.ac.uk (E.S.);; 2Jules-Gonin Eye Hospital, Lausanne University, 1011 Lausanne, Switzerland; 3Scheie Eye Institute, University of Pennsylvania, Philadelphia, PA 19104, USAkdine@pennmedicine.upenn.edu (K.D.); 4Imperial College London Ophthalmology Research Group, London NW1 5QH, UK; 5Western Eye Hospital, London NW1 5QH, UK

**Keywords:** resveratrol, neuroprotection, demyelinating disease

## Abstract

Purpose: Resveratrol is a natural polyphenol which has a very low bioavailability but whose antioxidant, anti-inflammatory and anti-apoptotic properties may have therapeutic potential for the treatment of neurodegenerative diseases such as multiple sclerosis (MS). Previously, we reported the oral administration of resveratrol nanoparticles (RNs) elicited a neuroprotective effect in an experimental autoimmune encephalomyelitis (EAE) mouse model of MS, at significantly lower doses than unconjugated resveratrol (RSV) due to enhanced bioavailability. Furthermore, we demonstrated that the intranasal administration of a cell-derived secretome-based therapy at low concentrations leads to the selective neuroprotection of the optic nerve in EAE mice. The current study sought to assess the potential selective efficacy of lower concentrations of intranasal RNs for attenuating optic nerve damage in EAE mice. Methods: EAE mice received either a daily intranasal vehicle, RNs or unconjugated resveratrol (RSV) for a period of thirty days beginning on the day of EAE induction. Mice were assessed daily for limb paralysis and weekly for visual function using the optokinetic response (OKR) by observers masked to treatment regimes. After sacrifice at day 30, spinal cords and optic nerves were stained to assess inflammation and demyelination, and retinas were immunostained to quantify retinal ganglion cell (RGC) survival. Results: Intranasal RNs significantly increased RGC survival at half the dose previously shown to be required when given orally, reducing the risk of systemic side effects associated with prolonged use. Both intranasal RSV and RN therapies enhanced RGC survival trends, however, only the effects of intranasal RNs were significant. RGC loss was prevented even in the presence of inflammatory and demyelinating changes induced by EAE in optic nerves. Conclusions: The intranasal administration of RNs is able to reduce RGC loss independent of the inflammatory and demyelinating effects on the optic nerve and the spinal cord. The concentration of RNs needed to achieve neuroprotection is lower than previously demonstrated with oral administration, suggesting intranasal drug delivery combined with nanoparticle conjugation warrants further exploration as a potential neuroprotective strategy for the treatment of optic neuritis, alone as well as in combination with glucocorticoids.

## 1. Introduction

Multiple sclerosis (MS) is a demyelinating inflammatory and neurodegenerative disease of the central nervous system (CNS). A common manifestation is optic neuritis associated with permanent vision loss from retinal ganglion cell (RGC) damage [1]. To characterize the disease process, one of the most common models used in research is the experimental autoimmune encephalomyelitis (EAE) mouse model. Present immunomodulatory treatments for MS seek to reduce inflammation caused by this condition, with limited effects on RGC neurodegeneration [1]. Nonetheless, neurodegenerative processes play an important role in the disability caused by this disease (blindness, paralysis) and represent a large social and economic burden.

The antioxidant, anti-apoptotic and anti-inflammatory properties of resveratrol are well documented and have been suggested to have a protective effect against neurodegeneration in glaucoma and other neurodegenerative diseases [2,3,4]. In the EAE mouse model, our group has previously shown that daily oral doses of 16.9 mg/kg resveratrol nanoparticles (RNs) were able to reduce RGC loss and reduce the severity of neurological dysfunction at a concentration six times lower than the minimum currently required of a 100 mg/kg oral dose of unconjugated resveratrol (RSV) diluted in phosphate-buffered saline (PBS) [5]. Oral RNs and RSV did not reduce inflammation, suggesting that they are capable of mediating neuroprotective effects independently of anti-inflammatory effects and may therefore provide synergistic mechanisms of action when administered in conjunction with current generation therapies [5].

Recent studies suggest the intranasal administration of a cell-derived secretome therapy containing numerous proteins at pg/mL to ng/mL concentrations leads to a selective accumulation of these agents in the eye and optic nerve, leading to RGC neuroprotection without modulating spinal cord disease [6]. The intranasal route of drug administration has received renewed interest in recent years as it typically requires lower doses than oral administration for CNS delivery [7], thus reducing the risk of systemic side effects [8].

In this study, we hypothesized that the intranasal administration of RNs will reduce the dose of resveratrol required to inhibit RGC loss, potentially increasing the neuroprotective efficacy of this therapy while reducing the risk of systemic side effects. Here, we examined whether our previously described RN formulation can prevent RGC loss at lower doses than previously reported for oral administration to selectively prevent RGC loss in EAE mice. The clinical manifestations in EAE as well as spinal cord and optic nerve inflammation and demyelination were also assessed to confirm whether neuroprotective effects occur independent of ongoing inflammation and demyelination after this route of administration.

## 2. Results

### 2.1. Intranasally Administered Resveratrol Nanoparticles Reduce RGC Loss

Mice were immunized with the MOG antigen on day 0 to induce EAE disease, and control mice were sham-immunized with an adjuvant alone (n = 5 non-EAE control mice). Mice were treated intranasally daily for 30 days with RNs, an equivalent vehicle (empty nanoparticles) or RSV. After one month of treatment, retinas were isolated and immunostained with Brn3a antibodies. Brn3a+ cells were counted by a masked investigator. The intranasal administration of 8.44 mg/kg RNs (n = 6 mice) resulted in a higher RGC density than the vehicle (n = 6 mice) (3569 ± 241 vs. 2280 ± 330 cells/1.56 mm^2^, *p* < 0.05) as shown in Figure 1, and treatment with intranasal 1.27 mg/kg RNs (n = 6 mice) showed a trend toward increased RGC survival (3070 ± 426 cells/1.56 mm^2^). Interestingly, 8.44 mg/kg RSV (n = 5 mice) also showed a trend towards neuroprotection with a non-significant higher RGC density compared to the vehicle (3072 ± 227 vs. 2280 ± 330 cells/1.56 mm^2^, *p* > 0.05), suggesting intranasally administered resveratrol may reach the retina without nanoparticle formulation.

### 2.2. RNs Do Not Significantly Alter Optic Nerve Inflammation and Demyelination or Visual Function

Optic nerves were isolated and stained to assess inflammation and demyelination. The intranasal administration of 8.44 mg/kg RNs resulted in just a trend towards inflammation reduction in the optic nerve compared to the 8.44 mg/kg vehicle as measured on a 0–4-point grading scale, but this was not significant (0.6 ± 0.1 vs. 1.2 ± 0.2 inflammation score, *p* > 0.05), as shown in Figure 2F. Additionally, 8.44 mg/kg RSV induced a similar non-significant trend in optic nerve inflammation as 8.44 mg/kg RNs (0.7 ± 0.2 vs. 0.6 ± 0.1 inflammation score, *p* > 0.05). Non-EAE control mice did not have detectable optic nerve inflammation, whereas EAE mice treated with the 8.44 mg/kg vehicle or 1.27 mg/kg RNs had significant inflammation as compared with controls (*p* < 0.001). A quantitative measurement of H&E staining also showed no significant difference between treated and untreated mice (Figure 2G).

The intranasal administration of 8.44 mg/kg RNs also showed a non-significant trend towards demyelination reduction in the optic nerve compared to the vehicle (0.6 ± 0.2 vs. 1.3 ± 0.4 demyelination score, *p* > 0.05), as shown in Figure 2M. Similar to optic nerve inflammation, this trend was also present with the administration of 8.44 mg/kg RSV compared to the vehicle (0.8 ± 0.2 vs. 1.3 ± 0.4 demyelination score, *p* > 0.05), and the measurement of the LFB staining intensity confirmed no significant differences between treated and untreated mice (Figure 2N).

Visual function was assessed using the OKR before immunization and once a week after immunization. The intranasal administration of RNs did not significantly alter the progressive decline in vision compared to the vehicle or RSV groups (Figure 3A). On day 28, 1.27 mg/kg RNs, 8.44 mg/kg RNs and 8.44 mg/kg RSV showed a small trend toward higher visual responses compared to the 8.44 mg/kg vehicle, but this was not significant (0.13 ± 0.04, 0.12 ± 0.03 and 0.17 ± 0.06 vs. 0.08 ± 0.03 OKR scores, *p* > 0.05), as shown in Figure 3B.

### 2.3. Similar Degree of EAE Clinical Disease and Spinal Cord Pathology Was Induced in All Treatment Groups

Clinical manifestations of the EAE ascending paralytic disease were measured daily in each mouse using the clinical score described above. EAE mice treated with the daily intranasal administration of 1.27 or 8.44 mg/kg RNs had equivalent EAE scores as compared with vehicle-treated EAE mice (2.0 ± 0.6 and 2.4 ± 0.2 vs. 2.3 ± 0.6 EAE score, *p* > 0.05, Figure 4). Similarly, RSV did not prevent the development of EAE disease.

After 30 days of treatment, isolated spinal cords were stained to assess inflammation and demyelination. Figure 5A–E show representative images of spinal cords with the presence or absence of inflammatory cells. EAE mice that received 1.27 or 8.44 mg/kg RNs intranasally showed a trend towards reduced spinal cord inflammation as assessed by masked investigators, but this was not significant compared to the vehicle (1.3 ± 0.5 or 1.3 ± 0.4 vs. 2.0 ± 0.4, *p* > 0.05), and mice treated with 8.44 mg/kg RSV showed equivalent inflammation as compared to 8.44 mg/kg RNs (1.2 ± 0.5 vs. 1.3 ± 0.4, *p* > 0.05) (Figure 5F). A quantitative measurement of H&E staining also showed no significant difference between treated and untreated mice (Figure 5G). Non-EAE control mice did not show inflammation in their spinal cords.

The intranasal administration of 1.27 or 8.44 mg/kg RNs did not show any reduction in spinal cord demyelination compared to vehicle treatment in EAE mice (1.4 ± 0.4 or 1.4 ± 0.4 vs. 1.2 ± 0.5, *p* > 0.05, Figure 5M), as illustrated by Figure 5H–N.

## 3. Discussion

The results of the current study demonstrate that RNs given intranasally to EAE mice were able to significantly reduce the RGC loss that occurs in EAE optic neuritis. Similar RGC neuroprotective effects were previously reported with the daily oral administration of this same RN formulation [5]. Of note, significant protective effects were observed here despite the maximum amount of RNs that could be administered intranasally (8.44 mg/kg, based on volume limitations) being only half of the 16.9 kg/mg dose previously shown to be required to prevent RGC loss with oral treatment [5]. Significantly lower doses of intranasal RNs (1.27 mg/kg), and equal doses of intranasal RSV, also showed strong trends toward reducing RGC, although not significant. Overall, the results suggest the intranasal administration of RNs warrants further study as a potential neuroprotective therapy to reduce the RGC loss induced by optic neuritis.

In agreement with our previous study [5], intranasal RN administration increased RGC survival without a reduction in inflammation and demyelination in the optic nerve and spinal cord, suggesting that neuronal survival can be promoted independent of, and despite, ongoing inflammatory demyelinating disease mechanisms. Interestingly, alternative treatment designed to upregulate the SIRT1 deacetylase via gene therapy is also neuroprotective despite having a limited effect on inflammation [9]. In contrast, several studies have shown potential anti-inflammatory properties of resveratrol in neurological-induced diseases leading to a preserved blood–brain barrier [10], by inhibiting the NF-κB signaling pathway [11] or the secretion of some cytokines directly [12] or even by promoting remyelination at 250 mg/kg resveratrol given orally [13]. As the concentration used in the current study is less than 5% of what is reported in the cited study [13], and the administration route is intranasal, it could explain why our nanoparticles did not show anti-inflammatory and anti-demyelination properties. Of note, current studies were designed to identify the minimal dosing necessary to illicit neuroprotective effects and demonstrate efficacy even in the presence of ongoing inflammation and demyelination, although future studies could be considered to determine whether higher intranasal doses might exert additional effects. In addition, similarities to prior EAE studies [5,9] using an alternative treatment delivery, which also showed that RGC survival was increased without a significant reduction in optic nerve inflammation, is more suggestive of disease-selective effects of resveratrol and other SIRT1-based therapies promoting neuroprotection without anti-inflammatory effects in optic neuritis. Together, the results of EAE optic neuritis studies suggest that intranasal RNs may provide an additive therapy that can be explored in combination with high-dose corticosteroids that are currently used to treat the acute inflammation in optic neuritis but fail to prevent RGC loss and the corresponding permanent visual changes [14]. High-dose corticosteroids are able to prevent inflammation by inhibiting microglial cells to produce inflammatory cytokines [15].

Consistent with observed ongoing inflammation, we also showed that the neuroprotective effects of intranasal RNs occur without reducing the severity of EAE neurological dysfunction, in the form of ascending paralysis or visual dysfunction. This is not unexpected, as active inflammation and demyelination may limit the function of surviving neurons but might allow for functional recovery after inflammation subsides, something that could be explored in future studies. The finding that the EAE ascending paralysis and spinal cord pathology was similar in all EAE groups in the current study is reassuring, as it demonstrates that EAE was successfully induced to a similar degree in each group.

To our knowledge, no prior studies using intranasal resveratrol in an animal model of MS have shown a neuroprotective effect to date. However, studies using other potential neuroprotective therapies delivered intranasally showed a reduction in RGC loss. For example, we treated the same animal model with intranasal ST266, a progenitor cell-produced therapy containing numerous growth factors and cytokines, and found significant RGC neuroprotective as well as optic nerve anti-inflammatory effects with this complex therapy [6,16], further supporting the idea that combined anti-inflammatory and neuroprotective therapies may be useful to attenuate optic neuritis. Zhou et al. also showed that intranasal erythropoietin improves both cognitive and visual function in a cerebral ischemia rodent model [17]. Few, if any, other groups have examined intranasal drug delivery for treating optic neuropathies; however, intranasal delivery has been shown to bypass the blood–brain barrier for targeting central nervous system disease [18,19], including studies showing the potential effects of intranasal RSV on Alzheimer’s disease [20].

Interestingly, the intranasal administration of 8.44 mg/kg RSV showed a trend towards a neuroprotective effect, despite prior studies showing an oral administration of at least 100 mg/kg RSV was needed to achieve a reduction in RGC loss [5]. A possible explanation for this might be that the nasal route bypasses the liver metabolism and the blood–brain barrier thus the unformulated drug can directly reach the brain, the optic nerve and the vitreous [6,21,22]. One potential hindrance to using the nasal route is the nasal mucosa, which contains p-glycoprotein, the multidrug resistance-associated protein (MRP1), metabolizing enzymes and mucociliary clearance mechanisms which reduce the amount of drug reaching the CNS [22]. The pathways used to reach the CNS remains unclear, but studies have shown that the olfactory or the trigeminal pathways might be one way to the CNS, as well as entrance into the system circulation and distribution through blood vessels such as the ophthalmic artery [22]. The intranasal route might allow for more frequent dosing, reduce systemic complications and allow for rapid translation to the clinic [6]. However, the translation in humans might be more challenging as their olfactory region covers less nasal cavity than rodents, hence reducing the amount able to penetrate the CNS [23]. The intranasal administration of RNs reduces RGC loss in an animal model of multiple sclerosis at lower concentrations than previously found to be required with oral treatment, and the effects are exerted independent of ongoing inflammatory demyelination. Although resveratrol has a low solubility in water, unconjugated resveratrol also showed a trend towards reducing RGC loss when given intranasally. The results support further studies to evaluate RNs as a potential therapy for optic neuritis, as well as a possible future examination of whether a specific formulation of resveratrol might not be necessary when administered intranasally, a route bypassing liver metabolism.

## 4. Material and Methods

### 4.1. Resveratrol Nanoparticle Preparation

Resveratrol nanoparticles (RNs) consisting of a particle size below 20 nm and stability over 3 months were formulated as described previously [5]. Briefly, resveratrol was encapsulated in D-α-tocopherol polyethylene glycol 1000 succinate (TPGS) and Kolliphor HS15 (Solutol) using a thin rehydration technique with a rotatory evaporator. Their encapsulation efficiency was assessed using spectroscopic techniques as described previously [5]. RNs were diluted in DMSO and their absorbance measured at 328 nm. The Micellar Drug Concentration (MDC) was calculated using Beer–Lambert’s law [24]: MDC (mg/mL) = A/Ɛ·M_w_·DF, where A is the absorbance of resveratrol at 328 nm, Ɛ the extinction coefficient (31,683 L·mol^−1^·cm^−1^), M_w_ the molecular weight of resveratrol (228.25 g/mol) and DF the dilution factor in DMSO (usually 1500). These nanoparticles were lyophilized using a freeze-drier as previously described [5].

### 4.2. Animal Handling

Animal experiments were approved by the Institutional Animal Care and Use Committee at the University of Pennsylvania and performed in accordance with the ARVO Statement for the Use of Animals in Ophthalmic and Vision Research and with the ARRIVE guidelines. C57BL/6J 6-week-old female mice (Jackson Laboratory, Bar Harbor, ME, USA) were housed in an air-conditioned, 21 °C environment with a 12 h light–dark cycle where food and water were available *ad libitum*.

### 4.3. Experimental Autoimmune Encephalomyelitis Mouse Model

EAE was induced as described previously [6,16]. Mice were anesthetized with an intraperitoneal injection of a solution containing 0.2 mL of 10 mg/mL ketamine (Sigma, St. Louis, MO, USA) and 1 mg/mL xylazine (Sigma). A total of 300 µg of MOG peptide 35–55 emulsified in complete Freund’s adjuvant (CFA; BD Difco, Franklin Lakes, NJ, USA) with 2.5 mg/mL mycobacterium tuberculosis (Difco) was subcutaneously injected at different sites on their back (2 doses of 150 µg each) to immunize them. Control, non-EAE mice (n = 5) were injected with an equivalent volume of PBS (1X phosphate-buffered saline, PH 7.4; Corning, Corning, NY, USA) diluted in CFA. A total of 200 ng of pertussis toxin (List Biological, Campbell, CA, USA) was injected intraperitoneally in control and immunized mice at immunization and 2 days post-immunization.

EAE ascending paralytic disease was scored by a masked observer using the following scale: 0 = unaffected; 0.5 = partial tail paralysis; 1 = full tail paralysis or waddling gait; 1.5 = partial tail paralysis and waddling gait; 2 = full tail paralysis and waddling gait; 2.5 = one hindlimb partially paralyzed; 3 = one hindlimb fully paralyzed; 3.5 = full paralysis of one hindlimb and partial of the other; 4 = full paralysis of both hindlimbs; 4.5 = moribund state; and 5 = death.

### 4.4. Intranasal Resveratrol Treatment

A freeze-dried resveratrol formulation comprising TPGS/Solutol (RNs), freeze-dried empty nanoparticles (TPGS/Solutol only = vehicle) and unconjugated resveratrol diluted in PBS (RSV) were used. Freeze-dried samples were hydrated with milli-Q^®^ water 30 min before use. EAE mice received intranasal treatment daily for 30 days as follows: 8.44 mg/kg vehicle (n = 6), 1.27 mg/kg RNs (n = 6), 8.44 mg/kg RNs (n = 6) and 8.44 mg/kg RSV (n = 5). The highest dose tested (8.44 mg/kg) was based on the maximum amount of RNs that could be solubilized in a volume of 20 µL, the previously reported largest volume that could be placed in the nares without excess fluid draining out of the nose [25], and represents half of the previously reported dose of oral RNs (16.9 mg/kg) required to prevent RGC loss in EAE mice [5].

### 4.5. Measurement of Optokinetic Response (OKR)

The OKR was used to approximate the visual function of mice. OptoMotry software and an apparatus (Cerebral Mechanics Inc., Medicine Hat, AB, Canada) were used to measure the OKR as before [6]. Mice were unrestrained on a platform in a closed dark chamber containing four screens and a camera used to detect if mice track a 100% contrast grating with varying spatial frequency starting at 0.042 cycles/degree. Data are presented as cycles/degree.

### 4.6. Optic Nerve and Spinal Cord Inflammation and Demyelination Grading

Optic nerves and spinal cords were isolated at the time of sacrifice and fixed in 4% paraformaldehyde (PFA). Next, they were embedded in paraffin and cut into 5 µm sections using a microtome (longitudinally for the optic nerve and axially for the spinal cord). The sections used for inflammation grading were stained with H&E whose gross cellularity correlates with macrophage staining [16]. Spinal cords were transected in 5–6 sections/mouse at multiple levels which were embedded in the same paraffin block to ensure that all spinal cord levels were included on each slide for staining and quantification. The following 0–3-point scale was used by masked investigators viewing all spinal cord levels as previously described [26]: 0 = no inflammation, 1 = mild inflammation (less than 5 small foci of white matter inflammatory cell infiltration), 2 = moderate inflammation (5–9 small foci of inflammation or 1–2 large areas of inflammation), and 3 = severe inflammation (more than 10 small foci or more than 2 large areas of inflammation). Optic nerve grading was assessed using a 0–4-point scale as before [26]: 0 = no infiltration, 1 = mild cellular infiltration (focal inflammation involving less than 25% of the entire length of the optic nerve), 2 = moderate infiltration (25% to 50% of optic nerve involved), 3 = severe infiltration (50 to 75% involved), and 4 = massive infiltration (>75% involved).

For demyelination, the entire length of the optic nerve and transverse spinal cord sections were stained with Luxol Fast Blue (LFB; Sigma-Aldrich, Kent, UK) and assessed by masked investigators using previously described ratings [26]. For spinal cords, the grading was as follows: 0 = no demyelination, 1 = rare foci of demyelination, 2 = a few areas of demyelination, and 3 = large confluent areas of demyelination. For optic nerves, the grading was as follows: 0 = no demyelination, 1 = scattered foci of demyelination, 2 = prominent foci of demyelination, and 3 = large (confluent) areas of demyelination.

In addition to the manual grading of both inflammation and demyelination, algorithms were developed by an operator masked to tissue treatment identities to identify the labeling extent with the following processes. ImageJ FIJI color deconvolution with appropriate vectors [27] was applied to each image. Shanbhag thresholding was then applied to H&E-stained optic nerve and spinal cord slides to generate masks of the complete tissue area extent [28]. Otsu thresholding was used for spinal cords stained with LFB [29] and Yen thresholding for optic nerves stained with LFB [30]. A further thresholding of the labeled area was used to measure the area of the labeling extent. Graphs of the labeling area were plotted for a random selection of representative images.

### 4.7. Quantification of RGC Survival

After sacrifice, retinas were harvested, immunostained for Brn3a and quantified as before [5]. Briefly, retinas were washed and permeabilized in PBS + 0.5% Triton x-100 (Thermo Fisher, Waltham, MA, USA) three times before freezing at −80 °C, washed again and incubated overnight at 4 °C with a 1:2000 diluted rabbit anti-Brn3a antibody (SC-411 003, Synaptic Systems, Göttingen, Germany) in blocking buffer (PBS, 2% BSA, 2% Triton x-100). The next day, retinas were washed before incubation with a 1:5000 donkey anti-rabbit secondary antibody (A21206, Alexa Fluor 488, Invitrogen, Paisley, OR, USA) diluted in blocking buffer for two hours. Next, retinas were washed four times in 1xPBS before being mounted on glass slides with vectashield (Vector Laboratories, Newark, CA, USA). RGCs were imaged at 20× magnification with a fluorescence microscope in 12 standard fields: 1/6, 3/6 and 5/6 of the retinal radius in each quadrant and counted by a masked investigator with ImageJ analysis software.

### 4.8. Statistical Analysis

All data were analyzed using GraphPad Prism 6 (La Jolla, CA, USA) and statistical tests (Student’s *t*-test, Kruskal–Wallis test and ANOVA) were used as indicated. The results are displayed as the mean ± standard error of the mean (SEM). The results are statistically significant if *p* < 0.05.

## Figures and Tables

**Figure 1 ijms-25-04047-f001:**
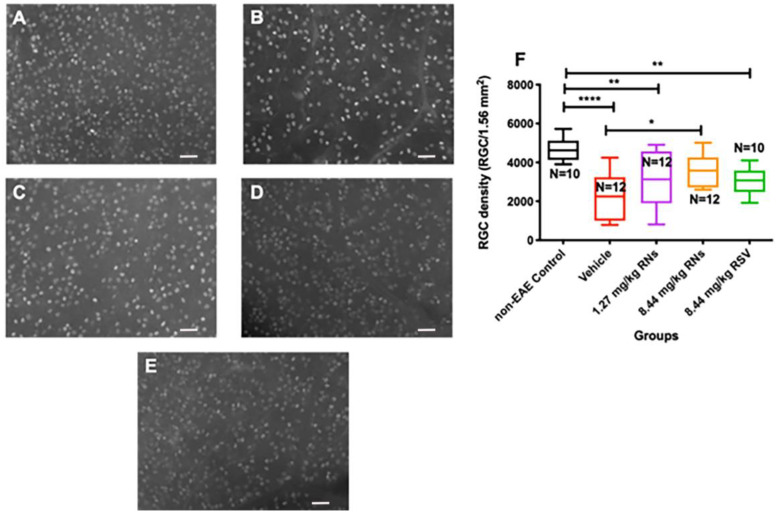
RGC survival after intranasal treatment with vehicle, RNs or RSV. Representative images of Brn3a+ cells in (**A**) control, (**B**) 8.44 mg/kg vehicle, (**C**) 1.27 mg/kg RNs, (**D**) 8.44 mg/kg RNs and (**E**) 8.44 mg/kg RSV. (**F**) RGC density was significantly increased with 8.44 mg/kg RN treatment compared to vehicle (one-way ANOVA with Tukey post-test, * *p* < 0.05). Control mice had higher RGC density than vehicle-, 1.27 mg/kg RN- and 8.44 mg/kg RSV-treated EAE groups (one-way ANOVA with Tukey post-test, ** *p* < 0.01, **** *p* < 0.0001). Scale bars 50 µm. RNs: resveratrol nanoparticles; RSV: unconjugated resveratrol.

**Figure 2 ijms-25-04047-f002:**
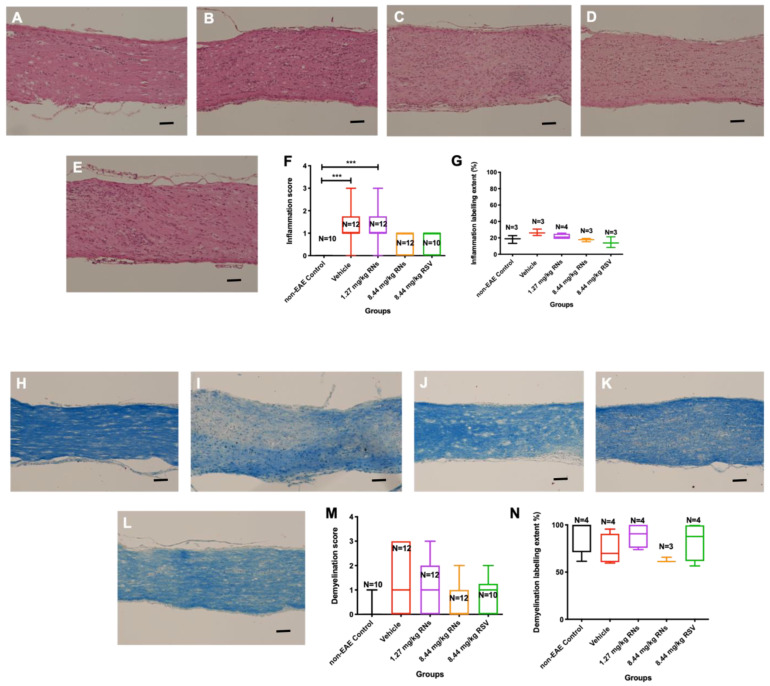
Optic nerve inflammation and demyelination after intranasal treatment with vehicle, RNs or RSV. Representative images of optic nerve inflammation in (**A**) control, (**B**) vehicle, (**C**) 1.27 mg/kg RNs, (**D**) 8.44 mg/kg RNs and (**E**) 8.44 mg/kg RSV. (**F**) Optic nerve inflammation score was not significantly reduced with RN or RSV treatment compared to equivalent vehicle (Kruskal–Wallis test with Dunn post-test, *p* > 0.05), while vehicle and 1.27 mg/kg RN groups had significantly higher inflammation score than non-EAE control group (*** *p* < 0.001). (**G**) Quantitative assessment of H&E staining intensity in randomly selected representative images also showed no differences between treated and untreated mice. Representative images of optic nerve longitudinal section demyelination in (**H**) control, (**I**) vehicle, (**J**) 1.27 mg/kg RNs, (**K**) 8.44 mg/kg RNs and (**L**) 8.44 mg/kg RSV. Optic nerve demyelination score was not reduced with RN or RSV treatment compared to equivalent vehicle (*p* > 0.05) as measured by qualitative scoring scale (**M**) or quantitative staining intensity (**N**). Scale bars 100 µm.

**Figure 3 ijms-25-04047-f003:**
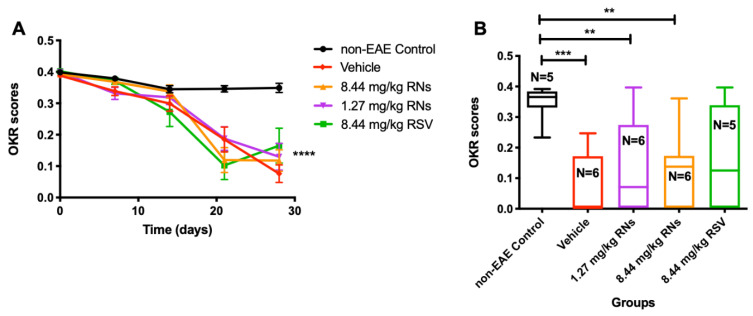
Visual function measured by OKR. (**A**) Visual function was reduced in EAE mice compared to non-EAE mice (two-way ANOVA with Tukey post-test, **** *p* < 0.0001) with or without RNs or RSV treatment. (**B**) On day 28, OKR score showed small non-significant trend toward higher scores in 1.27 mg/kg and 8.44 mg/kg RN- and 8.44 mg/kg RSV-treated EAE mice compared to vehicle-treated EAE mice (Kruskal–Wallis with Dunn post-test, *p* > 0.05). Control mice had higher OKR score than EAE mice treated with vehicle or RNs (Kruskal–Wallis with Dunn post-test, ** *p* < 0.01, *** *p* < 0.001).

**Figure 4 ijms-25-04047-f004:**
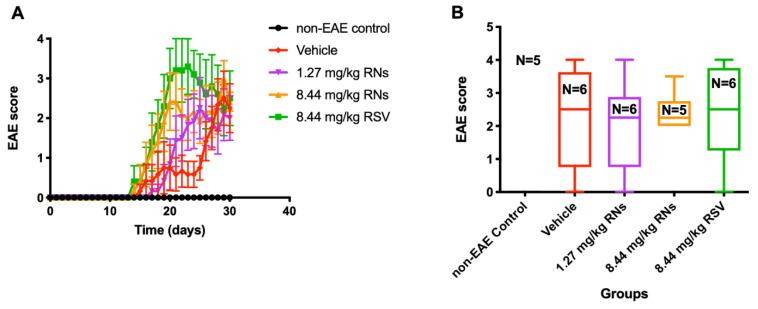
Clinical manifestation in EAE mice treated intranasally. (**A**) EAE score measured daily was similar in 8.44 mg/kg RNs’ group compared to vehicle (two-way repeated measures ANOVA with Tukey post-test, *p* > 0.05) (**B**) and remained similar at end of experiment on day 30 (Kruskal–Wallis test with Dunn post-test, *p* > 0.05).

**Figure 5 ijms-25-04047-f005:**
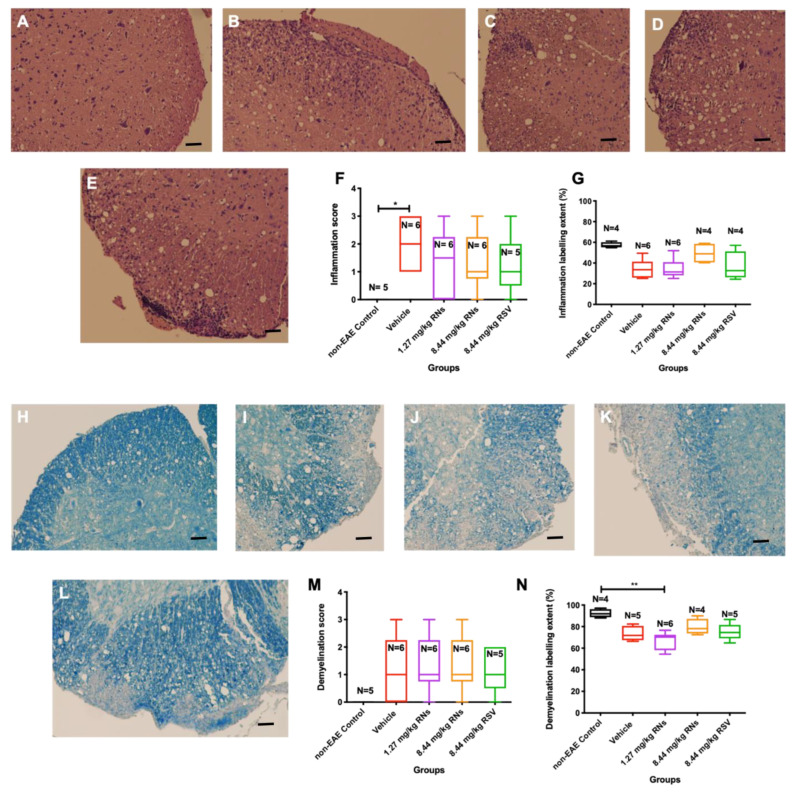
Spinal cord inflammation and demyelination after intranasal treatment with vehicle, RNs or RSV. Representative images of inflammation in H&E-stained transverse sections from multiple levels of spinal cord in (**A**) non-EAE control, (**B**) vehicle-, (**C**) 8.44 mg/kg RN-, (**D**) 1.27 mg/kg RN- and (**E**) 8.44 mg/kg RSV-treated EAE mice. (**F**) Significant spinal cord inflammation present in vehicle-treated EAE mice as compared to control mice (* *p* < 0.05) was not reduced with RN or RSV treatment (Kruskal–Wallis test with Dunn post-test, *p* > 0.05). (**G**) Quantitative assessment of H&E staining intensity in randomly selected representative images also showed no differences between treated and untreated mice. Representative images of demyelination in LFB-stained transverse sections from multiple levels of spinal cord in (**H**) non-EAE control, (**I**) vehicle-, (**J**) 1.27 mg/kg RN-, (**K**) 8.44 mg/kg RN- and (**L**) 8.44 mg/kg RSV-treated EAE mice. Spinal cord inflammation score was not reduced with RN or RSV treatment compared to vehicle treatment (*p* > 0.05), as measured by qualitative scoring scale (**M**) or quantitative staining intensity (**N**) (*p* > 0.05 for all comparisons except ** *p* < 0.01). Scale bars 50 µm.

## Data Availability

Data is contained within the article.

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
