# Peer review of "Intranasal Resveratrol Nanoparticles Enhance Neuroprotection in a Model of Multiple Sclerosis"

_ijms, 2024, doi:10.3390/ijms25074047_

Round 1

Reviewer 1 Report

Comments and Suggestions for Authors

The manuscript under consideration is a color image-based assessment of the neuroprotective effects of resveratrol nanoparticles in the mouse model of MS. It's an important area of research, and nanoparticle-mediated delivery of therapeutic agents is a promising modality. However, the presentation of results is not of sufficient quality in the manuscript. All images are missing scale bars, the size of image varies significantly in Figure 5, and the results are based on manual scoring by a "masked investigator". I suggest that the authors attempt an automated analysis, provide the full dataset and analysis outputs in the S.I., and resubmit the manuscript. Specific suggestions:

1. In Figure 1, automated RGC counting in FIJI is feasible after background subtraction and thresholding to generate a binary mask.

2. In Figures 2 and 5, automated analysis is also feasible after a) color deconvolution of H&E images and thresholding (or RGB thresholding) to count the inflammatory cells in FIJI and b) measurement of the intensity of Luxol Fast Blue (myelin) and total area of foci with lost myelin in FIJI. 

Author Response

Please see attached Response to Reviewers

Reviewer 2 Report

Comments and Suggestions for Authors

In this work, authors report on the use of intranasal resveratrol nanoparticles to enhance neuroprotection in a model of multiple sclerosis. They concluded that the intranasal administration of RNs reduces RGC loss in an animal model of multiple sclerosis at lower concentrations than previously found to be required with oral treatment and effects are exerted independent of ongoing inflammatory demyelination. It is a well written paper and the results presented are well supported by the data presented and the methodology used. I only have a few minor comments.

1. There are sone unnecessary gaps in the manuscripts that need to be corrected, line 176, 279 and 300.

2. There are far too many self-citations by the authors. I acknowledge the fact that they compare their results with previous ones produced by another administration method etc. and some of the experimental methods used are as described before, but from the 26 references at the end of the manuscript, 11 involve previous publications of the authors and I think some of them are not necessary and should be removed.

3. For some of the chemicals used like PBS and Triton x-100 there is no info regarding their purity or from where they were purchased. In addition, what was the concentration and the pH of PBS?

4. On page 10, line 316, there are no references given about other studies using other potential neuroprotective therapies delivered intranasally showing a reduction of RGC loss. There is only one mentioned afterwards by the same authors. I think more references should be added for neuroprotective therapies delivered intranasally that were developed by other research groups as well.

Author Response

(The authors gave the same response as above.)

Reviewer 3 Report

Comments and Suggestions for Authors

The current study uses intranasal resveratrol nanoparticles (RNs) for attenuating optic nerve damage in EAE mice. The concentration of RNs needed to achieve neuroprotection is lower than previously demonstrated with oral administration.

 Specific comments.

-Resveratrol nanoparticles (RNs) were formulated as described previously but information about the size of nanoparticles and their stability would improve the readability.

-The authors consider that the low resveratrol concentration (less than 5%) explains why nanoparticles did not show anti-inflammatory and anti-demyelination properties. It is somehow surprising to reach such a conclusion without any attempt to improve the resveratrol encapsulation rate. Additional experiments are required to improve the encapsulation yield by modifying the TPGS/HS15Kolliphor ratio and increasing the resveratrol initial concentration.

- Figure 2L- Intranasal administration of 8.44 mg/kg RNs showed a non-significant trend towards demyelination reduction in the optic nerve compared to vehicle. Again assays carried out at higher concentrations of encapsulated resveratrol are needed to reach such a conclusion.

Author Response

(The authors gave the same response as above.)

Round 2

Reviewer 1 Report

Comments and Suggestions for Authors

The authors addressed my concerns fully. 

Reviewer 3 Report

Comments and Suggestions for Authors

.